# The role of E-cadherin expression in the treatment of western undifferentiated early gastric cancer: Can a biological factor predict lymph node metastasis?

Gaetano Piccolo[1]*, Antonio Zanghì[1], Maria Di Vita[1], Pietro Bisagni[2], Francesca Lecchi[3], Andrea Cavallaro[1], Francesco Cardì[1], Emanuele Lo Menzo[4], Alessandro Cappellani[1]

1 Department of General Surgery and Medical-Surgical Specialty, University of Catania, Catania, Italy, 2 Department of Surgery, Ospedale Maggiore di Lodi, ASST di Lodi, Lodi, Italy, 3 General Surgery Residency Program, University of Milan, Milan, Italy, 4 Section of Minimally Invasive and Endoscopic Surgery, Cleveland Clinic Florida, Weston, Florida, United States of America

* gpiccolo1983@gmail.com

## Abstract

The use of endoscopic techniques to cure small sized, well differentiated early gastric cancer has been adopted worldwide. In the Eastern world, endoscopic resection is being increasingly utilized to treat small undifferentiated early gastric cancer according to the extended criteria proposed by the Japanese Gastric Cancer Associations. However, studies in the Western world reported in these tumors a rate of nodal metastasis ranging between 5% and 20%, that is higher of those observed in Eastern counterparts. A tool to predict the risk of nodal dissemination would be of great use to guide treatment toward endoscopic resection. In our study, we propose E-cadherin expression as a biological factor to predict lymph node involvement.

We retrospectively reviewed the E-cadherin (E-cad) expression profile of all histological specimens of undifferentiated early gastric cancer from two Oncologic Departments and compared it with several tumor characteristics. A total of 39 patients with early gastric cancer met the inclusion criteria, of which 16 (41%) pT1a, and 23 (58.9%) pT1b SM1. Thirty-two patients (82%) underwent subtotal gastrectomy, whereas total gastrectomy was performed in only seven cases (17.9%). Patients were divided into two groups: low E-cad expression (E-cad 0/1+, 10 patients) and high E-cad expression (E-cad 2+/3+, 29 patients) according to the immunohistochemical assay (ICH). On univariate analysis, we found an association between low E-cad expression and low grading tumor (p = 0.019), pure undifferentiated histotype (PU-type) (p = 0.014), and lymph node involvement (N+) (p < 0.001). The association between low E-cad expression and lymph node metastasis was confirmed by multivariate analysis (OR = 14.5, 95% CI 3.46–60.76, p < 0.001). The loss of expression of E-cad may be a simple biological factor to predict lymph nodes metastasis in patients with undifferentiated early gastric cancer. Additional larger prospective studies are necessary to confirm these findings.

**Data Availability Statement:** All data are contained within the paper.

**Funding:** The authors received no specific funding for this work.

**Competing interests:** No authors have competing interests.

## Introduction

Early gastric cancer (EGC) is defined as a gastric cancer (GC) confined to the mucosa (pT1a) or the submucosa (pT1b) irrespective of lymph node metastasis (LNM). It bears a more favorable prognosis after conventional gastrectomy with nodal dissection compared to advanced gastric cancer cases [1].

For small EGC, endoscopic treatment such as mucosal resection (EMR) and submucosal dissection (ESD) has been widely adopted as an alternative to conventional surgery, because it preserves gastric function and consequently leads to an improved quality of life [2,3]. Endoscopic resection of small EGC is a standard therapy in Asia (Japan, China, and Korea) and is being increasingly utilized in the USA and Europe.

The Japanese Gastric Cancer Associations guidelines [4] proposed two different sets of endoscopic resections: standard and expanded. Both procedures are considered curative when all of the following conditions are met: en-bloc resection, negative horizontal margin (HM0), negative vertical margin (VM0), and absence of lympho-vascular infiltration.

Recently, the National Comprehensive Cancer Network (NCCN) [5] endorsed the Japanese Gastric Cancer guidelines and proposed EMR/ESD for small-sized ($\leq$ 2 cm) well and moderately well differentiated EGC. However, these organ-sparing approaches do not involve lymph node dissection. The incidence of lymph node metastasis (LNM) in EGC ranges from 5% to 20% according to the tumor stage, size, grade of differentiation, and geographic area [2–3]. This significant variability of LNM between geographic areas could be secondary to the different biological behavior of these tumors.

The rate of nodal metastasis for tumor confined to the mucosa (pT1a) is significantly higher in Western countries [6] (5–20%) than in Eastern ones (not exceeding 5%) [7]. According to the USA national SEER database, LNM for low-grade T1a tumors was present in 1.7%, 4.1%, 4.5%, and 20% for 0–2cm, 2–3cm, 3–4cm, and $\geq$ 4 cm tumor size, respectively [6].

On the other hand, there are only case reports that describe nodal involvement in small undifferentiated Eastern EGC [8,9].

Thus, in order to safely perform conservative endoscopic resection in western undifferentiated EGC, the risk of concurrent nodal metastasis should be accurately investigated. We propose the use of E-cadherin expression as a possible early biological factor to predict lymph node involvement and to guide the selection of less invasive treatment modalities.

## Materials and methods

### Ethics statement

Patients were not required to give consent for this study, due to the retrospective nature of the study. All analyzed data was anonymized without identifiers. The study was reviewed and approved by the Institutional Review Board and by the Ethics Committee of the General Surgery Department of Catania.

### Study design

We retrospectively reviewed the medical records of all the patients who were treated for gastric cancer at two large referral institutions (the Department of Surgery of the University of Catania and the Department of Surgery of the Main Hospital of Lodi) between October 2015 and October 2019 All patients with undifferentiated pT1a or pT1b (SM1 < 500 μm from the muscularis mucosae) EGC were included in the study and their histological specimens were tested for E-cadherin expression profile. The primary endpoint was the evaluation of the prevalence of LNM and the correlation with the degree of E-cadherin expression. The overall surgical

outcomes were also analyzed and reported. Preoperative assessment included a complete medical history, physical examination, endoscopic ultrasound (EUS) to measure the depth of invasion, and upper gastrointestinal endoscopy with biopsies. Abdominal and chest CT scans were performed to assess the presence of local infiltration to adjacent organs, regional and distant nodal disease and lung, liver and/or peritoneal metastases. The goal of surgical procedure was a complete resection (R0) of the tumor.

Distal sub-total gastrectomy was performed in cases of tumors located in the lower and middle third of the stomach, if a proximal margin of at least 5 cm was feasible to achieve. Lymphadenectomy involved the systematic removal of perigastric lymph node stations (n˚ 1–7), and those along the celiac axis (n˚ 9), hepatic artery (n˚ 8a), splenic artery (n˚ 11p/d), and hepatoduodenal ligament (n˚ 12a). Lymph nodes at the splenic hilum were removed by splenectomy only when macroscopically involved.

The American Society of Anesthesiologists (ASA) score was used to stratify patients according to their perioperative risk. Tumor specimens were classified according to the Macroscopic Classification of the Japanese Gastric Cancer Association [4].

## Immunohistochemistry (IHC)

A representative paraffin block was obtained from each case. Immunohistochemistry (IHC) examination was performed using an automatic immune-stainer (DAKO OMNIS). Subsequently, the slides were incubated for one hour with the corresponding monoclonal antibody (clone 36B5). Each immunohistochemical staining was evaluated through a photomicroscope (Olympus®). Image acquisition was performed by Nano Zoomer-XR C12000 series (Hamamatsu Photonics®). The E-cadherin (E-cad) expression profile was stratified according to the grading system described by Chu et al. [10]:

- Absent (0): staining in fewer than 10% of tumor cells;

- Low (1+): weak staining in only 10%-50% of tumor cells;

- Low-intermediate (2+): moderate staining in 50%-75% of tumor cells;

- High (3+): strong staining of more than 75% of tumor cells.

## Statistical analysis

Statistical analysis was performed using the Statistical Package for the Social Sciences (SPSS) version 20.0. Data were represented as absolute frequency. For the univariate analysis, Mann-Whitney U tests were used. A p-value $<0.05$ was considered significant with confidence intervals (CI) of 95%. For the multivariate analysis, we used Cox-logistic regression analysis.

## Results

A total of 39 patients with early gastric cancer met the inclusion criteria, of which 16 (41%) pT1a, and 23 (58.9%) pT1b SM1. Thirty-two (82%) patients underwent subtotal gastrectomy, whereas total gastrectomy was performed in only seven cases (17.9%). Patient's characteristics are summarized in Table 1. The mean age was 62.7 years (± 9.2) with female predominance (59%).

EGC was classified according to the Macroscopic Classification of the Japanese Gastric Cancer Association [4]: the majority of tumors were Type-0-IIa (33.3%) (*superficial elevated*) or Type 0-III (30.8%) (*excavated*). Six cases were Type-0-I (15.4%) (*protruding*), four were

**Table 1. Patient's characteristics.**

| Patients | n/tot al (%) |
|---|---|
| **Sex** | |
| **Male** | 16/39 (41%) |
| **Female** | 23/39 (59%) |
| **Age (years)** | 62.7± 9.2 (range 45–75) |
| **Gastrectomy** | |
| **Subtotal** | 32/39 (82%) |
| **Total** | 7/39 (18%) |
| **ASA score** | |
| **1** | 1/39 (2.6%) |
| **2** | 11/39 (28.2%) |
| **3** | 25/39 (64.1%) |
| **4** | 2/39 (5.1%) |
| **Macroscopic Type** | |
| **Type-0- I** | 6/39 (15.4%) |
| **Type-0-IIa** | 13/39 (33.3%) |
| **Type-0- IIb** | 4/39 (10.2%) |
| **Type-0-IIc** | 4/39 (10.2%) |
| **Type-0- III** | 12/39 (30.8%) |
| **Histological type** | |
| **MU-Type** | 17/39 (43.6%) |
| **PU-Type** | 22/39 (56.4%) |
| **SRC** | 16/22 (72.7%) |
| **Poor** | 5/22 (22.7%) |
| **Muc** | 1/22 (4.5%) |
| **pT Stage** | |
| **pT1a** | 16/39 (41%) |
| **pT1b (SM1)** | 23/39 (60%) |
| **Grading** | |
| **G1** | 10/39 (25.6%) |
| **G2** | 18/39 (46.1%) |
| **G3** | 11/39 (28.2%) |
| **Ulcerative finding** | |
| **UL (+)** | 18/39 (46.1%) |
| **UL (-)** | 21/39 (53.8%) |
| **pN** | |
| **pN0** | 31/39 (79.5%) |
| **pN1** | 8/39 (20.5%) |

MU = mixed undifferentiated

PU = pure undifferentiated

SRC = signet-ring cell carcinoma

Poor = poorly solid adenocarcinoma

Muc = mucinous tumor

Type-0-IIb (10.2%) (*superficial flat*) and four Type-0-IIc (10.2%) (*superficial depressed*). Ulcerative findings (UL +) were present in 21/39 patients (53.8%).

Undifferentiated EGC included pure undifferentiated (PU-type; 56.4%) and predominantly or mixed undifferentiated cases (MU-type; 43.6%). Among PU-type tumors, there were 16

signet-ring cell carcinomas (SRC; 72.7%), five poorly solid or non-cohesive differentiated adenocarcinomas (poor; 22.7%) and one mucinous tumor (muc; 4.5%).

All patients underwent D2 lymphadenectomy and the mean number of lymph nodes retrieved was 15.47 (range 4–23). According to the 8[th] AJCC nodal involvement classification [11], our series included 31 (79.5%) pN0 cases (no regional lymph node metastasis) and eight (20.5%) pN1 cases (metastasis in one or two regional lymph nodes).

We analyzed the relationship between E-cad expression and some clinic-pathological features: histotype, depth of invasion, grading, tumor size and N status. All cases enrolled in our study were classified into two groups: low E-cadherin expression (E-cad 0/1+) and high E-cadherin expression (E-cad 2+/3+) (S1 Fig). On univariate analysis (Table 2), we found an association between low E-cadherin expression and low tumor grading (p = 0.019), pure undifferentiated histotype (PU-type) (p = 0.014) and lymph node involvement (N+) (p < 0.001). The association between low E-cadherin expression and lymph node metastasis (LNM) was confirmed by multivariate analysis (OR = 14.5, 95% CI 3.46–60.76, p < 0.001) (Table 3).

## Discussion

Gastric cancer (GC) incidence has decreased in Western countries due to the diffusion of eradication therapy for H. Pylori and the improvement in food preservation methods. However, GC is still the fifth solid cancer for frequency and the third cause of cancer-related death (over 934,000 new cases and 720,000 deaths per year) worldwide [12]. Early gastric cancer (EGC) is generally associated with a better prognosis, but nodal involvement has to be considered. Gastric cancer is classified in two different groups: differentiated and undifferentiated type. The former includes papillary adenocarcinoma (pap) and well and moderately differentiated tubular adenocarcinoma (tub1, tub2); the latter includes poorly-differentiated tubular adenocarcinoma, poorly differentiated adenocarcinoma, solid and non-solid type (por1, por2), signet-ring cell carcinoma (sig) and mucinous adenocarcinoma (muc) [13]. The risk of lymph node involvement is significantly higher for undifferentiated rather than differentiated tumors. Therefore, some cases of EGC may be characterized by unfavorable histological type, low degree of differentiation and high metastatic potential.

In order to obtain a more tailored treatment plan, an accurate research of early biological factors to predict lymph node metastasis is needed. Several markers have been studied, among which tumor size, depth of invasion, macroscopic and histological type are widely considered as risk factors for nodal involvement [14,15].

The metastatic capacity of cancer cells originates from the acquired ability to lose normal adhesion with adjacent structures (homing) and to spread through the lymphatic system or bloodstream with possible distant organ invasion. During the epithelial–mesenchymal transition (EMT), the loss of intercellular adhesions is likely to be the first step toward the metastatic phase. E-cadherin protein (endothelium calcium-dependent adhesion protein) plays a leading role in this process. It is a class of type 1 trans-membrane proteins that links catenins to form an E-cadherin/catenin complex which is linked to the actin cytoskeleton [16].

The use of E-cadherin (E-cad) expression as a molecular marker was analyzed in several studies leading to interesting but not definitive results.

In a retrospective study, Shun et al. [17] demonstrated a correlation between the loss of E-cad and the histological type, with abnormal expression being more frequent in diffuse-type tumors compared to intestinal-type (p < 0.0005). The authors also reported a correlation between abnormal expression of E-cad and a higher frequency of lymph node metastasis (LNM) (p < 0.05). Cai et al [18], in a large study on 162 patients, reported that the loss of E-cad together with lymphatic invasion of the primary tumor represented an independent risk

**Table 2. Univariate analysis between low E-cadherin expression (E-cad 0 / +1) and high E-cadherin expression (E-cad +2 / +3).**

| | N° Positivity to E- cad 0/1+ | N° Positivity to E-cad 2+/3+ | Mann-Whitney U-Test (p) |
|---|---|---|---|
| **Macroscopic Type** | | | p = 0.144 |
| **Type-0- I** | 2 | 4 | |
| **Type-0-IIa** | 3 | 10 | |
| **Type-0- IIb** | 1 | 3 | |
| **Type-0-IIc** | 0 | 4 | |
| **Type-0- III** | 4 | 8 | |
| **Histotype** | | | |
| | | | p = 0.014 |
| **MU-Type** | 1 | 16 | |
| **PU-Type** | 9 | 13 | |
| **Depth of invasion** | | | p = 0.122 |
| pT1a | 2 | 14 | |
| pT1b sm1 | 8 | 15 | |
| **Grading** | | | p = 0.019 |
| G1 | 1 | 9 | |
| G2 | 3 | 15 | |
| G3 | 6 | 5 | |
| **Size** | | | p = 0.415 |
| < 1 cm | 1 | 6 | |
| ≥ 1 to ≤ 2 cm | 3 | 12 | |
| >2 cm | 6 | 11 | |
| **N Status** | | | p < 0.001 |
| N0 | 2 | 29 | |
| N+ | 8 | 0 | |

MU = mixed undifferentiated

PU = pure undifferentiated

factor for LNM (p < 0.05). Additionally, in a recent meta-analysis the authors [19] found a statistical correlation between the E-cad expression and some clinical variables such as depth of invasion (p < 0.001), lymph node spread (p < 0.001) and distant metastasis (p < 0.001), leading to a poor 5-year overall survival (p < 0.001).

The loss of E-cad expression can be caused by several mechanisms [20–24]. The mutation or deletion of CDH1 (E-cadherin gene) can be found in some hereditary gastric cancers [25] but it is also common in familiar cases [26]. On the other hand, in sporadic GC, somatic mutation of CDH1 is extremely rare and is more frequent in the diffuse phenotype [27]. In many cases there are no structural mutations, but the loss of E-cad is due to epigenetic alterations such as promoter hypermethylation or activation of transcriptional repressors.

MicroRNAs (non-coding RNAs) modulate CDH1 through the regulation of its transcription factors (EZH2, ZEB1 and ZEB2). Rossi T et al. [28] investigated the impact of microRNAs

**Table 3. Multivariate analysis between low E-cadherin expression (E-cad 0 / +1) and high E-cadherin expression (E-cad +2 / +3).**

| | N° Positivity to E- cad 0/+1 | N° Positivity to E-cad +2/+3 | OR | (95% CI) | p |
|---|---|---|---|---|---|
| **N Status** | | | 14.5 | 3.46–60.76 | p < 0.001 |
| N0 | 2 | 29 | | | |
| N+ | 8 | 0 | | | |

on intestinal gastric cancer (IGCs) compared to normal tissues. They found a significant downregulation of miR-101, miR-26b and miR-200 in respectively 57.6%, 51.5%, and 51.5% of IGCs. Moreover, miR-200 expression seems to act as tumor suppressor, correlating with lower tumor grade. Twist, Snail and Slug are transcriptional repressors of CHD1, inhibiting the expression of genes containing E-boxes in the promoter regions. Chen et al. [29] revealed Snail expression in metastatic lymph nodes. When tumor cells transfer from metastatic lymph nodes to another lymph node, Snail expression seems to be generally over-expressed. We believe that the new concept of the Snail switch, which is the positive-to-negative conversion of the Snail status in metastatic lymph nodes, may explain the loss of E-cad in the primary tumor. According to our results, E-cad expression may be a simple biological factor to predict LNM in patients with undifferentiated EGC. However, our study is subjected to many limitations, the most important of which is the relatively small series of cases enrolled. Larger studies are needed to validate this theory.

## Conclusion

Detection of E-cad on undifferentiated EGC could be a feasible method to predict which patients should undergo endoscopic resection and which ones should be submitted to surgery with extended lymphadenectomy.

## Supporting information

**S1 Fig. The E-cadherin (E-cad) expression by immunohistochemical assays (ICH).** (PDF)

## Author Contributions

**Conceptualization:** Gaetano Piccolo.

**Data curation:** Gaetano Piccolo, Maria Di Vita.

**Investigation:** Gaetano Piccolo.

**Supervision:** Antonio Zanghì, Pietro Bisagni, Francesca Lecchi.

**Validation:** Alessandro Cappellani.

**Visualization:** Andrea Cavallaro, Francesco Cardì.

**Writing – original draft:** Gaetano Piccolo, Emanuele Lo Menzo.

**Writing – review & editing:** Gaetano Piccolo, Emanuele Lo Menzo.

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
