## [Decision Letter · Decision Letter 0]

17 Mar 2020

PONE-D-20-01711

The role of E-cadherin expression in the treatment of western undifferentiated early gastric cancer: can a biological factor predict lymph node metastasis?

PLOS ONE

Dear Dr Piccolo,

Thank you for submitting your manuscript to PLOS ONE. After careful consideration, we feel that it has merit but does not fully meet PLOS ONE’s publication criteria as it currently stands.

The association between low E-cad expression and lymph nodes metastasis is not new but of interest and study to confirmed this point is of interest. However, authors must refer to prevalence and not to incidence of nodal involvement. Moreover, a correlation of macroscopic aspect of early gastric cancer with E-cad expression could improve the study.

Therefore, we invite you to submit a revised version of the manuscript that addresses the points raised during the review process.

We would appreciate receiving your revised manuscript by May 01 2020 11:59PM. To enhance the reproducibility of your results, we recommend that if applicable you deposit your laboratory protocols in protocols.io, where a protocol can be assigned its own identifier (DOI) such that it can be cited independently in the future. For instructions see: http://journals.plos.org/plosone/s/submission-guidelines#loc-laboratory-protocols

We look forward to receiving your revised manuscript.

Kind regards,

Valli De Re, Ph.D.

Academic Editor

PLOS ONE

Journal Requirements:

3. At this time, we ask that you please provide additional information in your Methods section about the specific methodology used to conduct the immunohistochemical analysis in your study.

'No'

Please provide an amended Funding Statement that declares *all* the funding or sources of support received during this specific study (whether external or internal to your organization) as detailed online in our guide for authors at http://journals.plos.org/plosone/s/submit-nowPlease state what role the funders took in the study.  If any authors received a salary from any of your funders, please state which authors and which funder. If the funders had no role, please state: "The funders had no role in study design, data collection and analysis, decision to publish, or preparation of the manuscript."

5. Please include a caption for figure 1.

Reviewers' comments:

Reviewer's Responses to Questions

**Comments to the Author**

1. Is the manuscript technically sound, and do the data support the conclusions?

Reviewer #1: Yes

2. Has the statistical analysis been performed appropriately and rigorously? 

Reviewer #1: I Don't Know

3. Have the authors made all data underlying the findings in their manuscript fully available?

Reviewer #1: Yes

4. Is the manuscript presented in an intelligible fashion and written in standard English?

Reviewer #1: No

5. Review Comments to the Author

Reviewer #1: This is an interesting paper on an important topic. How to help to predict the presence of lymph node involvement in early gastric cancer. however authors must refer to prevalence and not to incidence of nodal involvment . Another interesting data is to correlate macroscopic aspect of early gastric cancer and E-cad. Paper must be revisad by native english

6. PLOS authors have the option to publish the peer review history of their article (what does this mean?). If published, this will include your full peer review and any attached files.

Reviewer #1: No

---

## [Author Response · Author response to Decision Letter 0]

4 Apr 2020

Journal Requirements:

We consulted PLOSE ONE’s online submission guidelines and carefully revised the manuscript style to meet the journal requirements.

2. We suggest you thoroughly copyedit your manuscript for language usage, spelling, and grammar.

The manuscript was kindly copyedited by natural English speaker: Emanuele Lo Menzo M.D, PhD, FACS, Associate Professor of Surgery, Department of General Surgery, Division of Minimally Invasive and Metabolic Surgery, Cleveland Clinic Florida, Weston, FL, USA.

3. At this time, we ask that you please provide additional information in your Methods section about the specific methodology used to conduct the immunohistochemical analysis in your study.

Thank you, we provided additional information about the IHC methodology used for the study and included it in the Methods section of the manuscript.

4. Please provide an amended Funding Statement that declares *all* the funding or sources of support received during this specific study.

The authors received no specific funding for this work. We modified our cover letter.

5. Please include a caption for figure 1. 

Figure 1 along with its caption was submitted as supporting information in the requested style and format. 

Review Comments to the Author

Reviewer #1: This is an interesting paper on an important topic. How to help to predict the presence of lymph node involvement in early gastric cancer. however authors must refer to prevalence and not to incidence of nodal involvment . Another interesting data is to correlate macroscopic aspect of early gastric cancer and E-cad. Paper must be revisad by native English

Thank you for the suggestion; We recognize our mistake in referring to incidence instead of prevalence and corrected it in the manuscript. The correlation between macroscopic features of EGC and E-Cad expression has been included in the revised manuscript. Paper has been revised by a native English speaker.

---

## [Editor Report · Decision Letter 1]

15 Apr 2020

The role of E-cadherin expression in the treatment of western undifferentiated early gastric cancer: can a biological factor predict lymph node metastasis?

PONE-D-20-01711R1

Dear Dr. Piccolo,

We are pleased to inform you that your manuscript has been judged scientifically suitable for publication and will be formally accepted for publication once it complies with all outstanding technical requirements.

With kind regards,

Valli De Re, Ph.D.

Academic Editor

PLOS ONE

Reviewers' comments:

authors make changes according to the request.

---

## [Editor Report · Acceptance letter]

17 Apr 2020

PONE-D-20-01711R1 

The role of E-cadherin expression in the treatment of western undifferentiated early gastric cancer: can a biological factor predict lymph node metastasis? 

Dear Dr. Piccolo:

I am pleased to inform you that your manuscript has been deemed suitable for publication in PLOS ONE. Congratulations! Your manuscript is now with our production department. 

With kind regards,

on behalf of

Dr. Valli De Re 

Academic Editor

PLOS ONE